# High-Temperature Oxidation Resistance and Molten Salt Corrosion Study of YSZ, CeYSZ, and YSZ/CeYSZ Thermal Barrier Coatings by Atmospheric Plasma Spraying

Rongbin Li [1,*], Yue Xing [2], Qianqian Li [2], Zhijun Cheng [2] and Linlin Guo [2]

1   School of Materials, Shanghai Dianji University, Shanghai 201306, China
2   School of Materials and Chemistry, Shanghai University of Technology, Shanghai 200093, China; 213353224@st.usst.edu.cn (Y.X.)
*   Correspondence: lirb@sdju.edu.cn; Tel.: +86-213-8223-8222

**Abstract:** Five thermal barrier coatings (TBCs), namely TBC-1 (YSZ), TBC-2 (CeYSZ), TBC-3 (YSZ:CeYSZ = 1:2), TBC-4 (YSZ:CeYSZ = 1:1), and TBC-5 (YSZ:CeYSZ = 2:1), were fabricated using the atmospheric plasma spraying (APS) method. Their oxidation behaviors at 1100 °C and corrosion resistance to molten salts ($V_2O_5$ + $Na_2SO_4$) at 900 °C were examined. After 100 h of oxidation, the thermally grown oxide layer (TGO) for YSZ primarily contained Cr and Ni oxides with significant internal fractures, presenting a continuous band-like $Al_2O_3$. In dual-ceramic configurations, an increase in CeYSZ thickness led to a rise in Al content and reduced Cr and Ni in TGO, with the surface fracture morphing into an internal porosity. Following salt corrosion, YSZ revealed rod-like $YVO_4$ and $m$-$ZrO_2$ as corrosion products, whereas CeYSZ displayed chain-structured $CeO_2$, $CeYO_4$, and $YVO_4$ combined with $m$-$ZrO_2$. YSZ coatings underwent notable phase transitions with evident densification, forming a corrosion layer of approximately 10 μm. Conversely, CeYSZ showed a limited phase change, retaining porosity without a distinguishable corrosion layer. As CeYSZ thickness increased from 100 μm to 200 μm in the dual-ceramic structure, salt penetration reduced. Evidently, the dense structure of CeYSZ heightened diffusion resistance against oxygen and corrosive salts, rendering superior oxidation and corrosion resistance over YSZ. By optimizing the thickness ratio between CeYSZ and YSZ, whilst retaining total ceramic layer thickness, the dual-ceramic TBC's resistance to high-temperature oxidation and corrosion can be enhanced.

**Keywords:** plasma spraying; thermal barrier coatings (TBC); high-temperature oxidation; molten salt corrosion

## 1. Introduction

With the continuous increase in the thrust-to-weight ratio of aero engines and the improvement of gas utilization efficiency, the future turbine inlet temperature can reach 2000 K. Presently, advanced nickel-based superalloys can elevate the pre-turbine inlet temperature up to 1323 K [1–3]. The design of blade cooling structures only slightly enhances the pre-turbine temperature by a few hundred Kelvin, but the thermal stresses induced by temperature gradients during the operation of the blade cooling system can affect the blade lifespan [4]. Consequently, the advancement of thermal barrier coating (TBC) technology is paramount [5,6].

Numerous scholars have extensively investigated the materials, structures, and failure modes of TBCs. Yttria-stabilized zirconia (YSZ), as the most classic TBC material, hosts the t′ phase (metastable tetragonal phase) in the coating, which remains stable at temperatures up to 1200 °C [7,8]. This stability enhances the coating's strain tolerance and resistance to thermal shock. Due to its exceptional thermophysical and mechanical properties, YSZ is considered the most widely used TBC material to date [9–11]. However, when the operating temperature of YSZ exceeds 1473 K, a transition occurs from the t′ phase to

the t phase (tetragonal phase) and c phase (cubic phase). During cooling, the t phase transforms into the m phase (monoclinic phase). This phase transformation is accompanied by a 3%–5% volume expansion, leading to the formation of cracks in the coating [12]. When YSZ coatings are exposed to high-temperature environments for prolonged periods, oxygen permeates the coating and forms a thermally grown oxide layer (TGO) on the bond coat. The thermal mismatch between the TGO and the coating and the growth stress of the TGO also contribute to coating failure [13–15]. In addition, the fuel used in gas turbines often contains sulfur, vanadium, and other elements. These elements react with the air, forming corrosive salts such as $V_2O_5$ and $Na_2SO_4$ [16]. These molten salts can penetrate the coating at high temperatures and react with the stabilizers in the coating. The phase changes during the reaction can lead to concentrated thermal stresses within the coating, triggering premature coating failure [17]. Therefore, modifying YSZ with rare earth oxides such as $La_2O_3$, $Sc_2O_3$, $Gd_2O_3$, $Yb_2O_3$, and $CeO_2$ has become a focal point of current research [18–23]. Studies indicate that doping YSZ with $CeO_2$ enhances the coating's thermal expansion coefficient, strain tolerance, and thermal cycle lifespan. The substitution of $Zr^{4+}$ by $Ce^{4+}$ reduces the coating's thermal conductivity. Moreover, CeYSZ coatings display no significant phase changes at 1573 K, indicating excellent high-temperature phase stability [24–26]. While doping with $CeO_2$ enhances the thermomechanical properties of the coating, specific studies on the influence of $CeO_2$ doping on the TGO growth behavior and molten salt corrosion of YSZ coatings are limited. The introduction of dual-ceramic-layer structures, composite coating structures, and functionally graded coating structures has paved the way for the development of superior TBCs. Among these, dual-ceramic-layer structures, which offer a higher thermal cycle lifespan than single-ceramic-layer TBCs and are simpler to produce, have garnered significant attention [27–30]. However, limited research has been conducted on the specific impact of ceramic layer thickness ratios in dual-ceramic-layer structures on coating performance [31].

Common coating preparation techniques include atmospheric plasma spraying (APS), electron beam physical vapor deposition (EB-PVD), and high-velocity oxygen fuel (HVOF) spraying. Atmospheric plasma spraying, due to its simplicity, low cost, rapid preparation speed, and high deposition efficiency, is widely used in the fabrication of thermal barrier coatings, wear-resistant coatings, restoration of corrosion-resistant coatings, and near-net shaping of advanced ceramics. It is particularly suitable for the preparation of thick coatings (300–3000 μm) [32–34]. Therefore, the preparation of all coatings in this study was carried out using the APS method.

To compare the resistance to molten salt corrosion and high-temperature oxidation performance between YSZ and CeYSZ coatings and to investigate the influence of the thickness ratio of the two coatings in the double-ceramic-layer structure on the overall performance of the coatings, YSZ and CeYSZ coatings were prepared using the APS method. The mechanisms of TGO growth and degradation, as well as corrosion behavior and mechanisms of both coatings, were analyzed. Three TBCs with different ceramic layer proportions were designed and fabricated. The impact of the thickness ratio between CeYSZ and YSZ in the double-ceramic-layer structure on the coating's resistance to high-temperature oxidation and corrosion was systematically explored and discussed.

## 2. Materials and Methods

### 2.1. Materials

The substrate material utilized was an Inconel 718 nickel-based superalloy disc with dimensions of Φ 25.4 mm × 5 mm. Prior to its use, the disc was subjected to polishing to guarantee a controlled surface roughness. The bond coat powder used for plasma spraying was commercial-grade NiCrAlY (Ni-22Cr-10Al-1.0Y, Amdry 962, Oerlikon Metco, Anaheim, CA, USA). The ceramic top coat powders employed were YSZ ($ZrO_2$-8$Y_2O_3$, Metco 204NS, Oerlikon Metco) and CeYSZ ($ZrO_2$-2.5$Y_2O_3$-10$CeO_2$, procured from China Liaoning Jinzhou Jinjiang Coating Materials Ltd.). Using atmospheric plasma spraying (APS) techniques, two single-ceramic-layer TBCs and three dual-ceramic-layer TBCs

with varying layer thickness ratios were fabricated. These were designated as TBC-1 (YSZ), TBC-2 (CeYSZ), TBC-3 (8YSZ:CeYSZ = 1:2), TBC-4 (8YSZ:CeYSZ = 1:1), and TBC-5 (8YSZ:CeYSZ = 2:1). The bond coat was approximately 100 μm thick, while the ceramic coat was around 300 μm in thickness. The motion system of the APS equipment is operated by an ABB robot, and the spray gun model is F4-MB 6 mm. The APS spraying parameters for both the bond coat and ceramic layers are detailed in Table 1.

**Table 1.** Spray parameters for the NiCrAlY bond coat and the ceramic top coats.

| Parameters | NiCrAlY Bond Coat | 8YSZ Top Coat | CeYSZ Top Coat |
|---|---|---|---|
| Current (A) | 600 | 600 | 600 |
| Primary Gas Flow (Ar, $L \cdot min^{-1}$) | 46 | 40 | 40 |
| Feed Rate ($g \cdot min^{-1}$) | 84 | 40 | 40 |
| Spray Distance (mm) | 90 | 100 | 100 |
| Secondary Gas Flow ($H_2$, $L \cdot min^{-1}$) | 6 | 8 | 8 |
| Step Size (mm) | 4 | 2 | 2 |
| Spray Gun Traverse Speed ($mm \cdot s^{-1}$) | 1000 | 1000 | 1000 |

*2.2. High-Temperature Oxidation Experiment*

Thermal oxidation tests were conducted on five coating structures using a muffle furnace. To prevent substrate detachment at elevated temperatures, the coatings along with their substrates were placed into crucibles before insertion into the muffle furnace. The samples were heated to a set temperature of 1100 °C and held for 100 h [35]. After the insulation process, the samples were cooled to room temperature, cut using grinding wheels into specimens, subjected to heat treatment with epoxy resin, and finally polished and finished with SiC sandpaper and 2.5 μm polishing paste.

*2.3. Molten Salt Corrosion Test*

A mixture of $Na_2SO_4$ and $V_2O_5$ (in a 50%:50% weight ratio) was employed as the corrosive salt [36]. This mixture was uniformly applied onto the coating surfaces at a rate of 10 mg/cm$^2$. To mitigate the edge effects of molten salt on the coating, a 2–3 mm gap was maintained between the corrosive salt and the coating's edge. Subsequently, samples were placed into the muffle furnace and held at 900 °C for 2 h. Upon completion, specimens were cooled down to ambient temperature.

*2.4. Microstructure and Phase Analysis*

X-ray diffraction (XRD) was utilized to analyze the phase composition of the sprayed powders, as-sprayed coatings, thermally oxidized coatings, and coatings post-molten salt corrosion. The microstructure of the sprayed powders, the surface and cross-sectional morphologies of the as-sprayed coatings, the cross-sectional morphology after oxidation, and the microstructure of samples after corrosion were characterized using scanning electron microscopy (SEM). The surface roughness of the coatings was measured with a Smart Proof 5 confocal microscope equipped with a rotational stage. Energy-dispersive spectroscopy (EDS) was employed to determine the elemental composition of the thermally grown oxide (TGO) in different coatings after high-temperature oxidation. Concurrently, EDS analysis was conducted on the corrosion products and cross-sectional elemental distribution of the samples post-corrosion. On the cross-section of the coating, six randomly selected regions were subjected to grayscale processing using Image J software. The ratio of the pore area to the total area of each region was calculated to determine the coating porosity. Multiple measurements were conducted, and the average value was recorded as the coating's average porosity. The surface roughness of the coating was measured using the Smart Proof 5 confocal microscope. An average of 10 measurements was taken as the coating's surface roughness value.

## 3. Results and Discussion

### 3.1. Characterization of As-Sprayed Coatings

3.1.1. Phase Analysis of the Coating

From the XRD spectra (Figure 1) of both powder and as-sprayed states, it is evident that both the YSZ and CeYSZ powders possess the m (monoclinic) phase of $ZrO_2$. Additionally, CeYSZ powder exhibits traces of the compound $Zr_{0.4}Ce_{0.6}O_2$. Conversely, the as-sprayed coatings of YSZ and CeYSZ predominantly showcase the t' (non-equilibrium tetragonal) phase. This observation can be attributed to the rapid cooling experienced by the molten or gaseous YSZ and CeYSZ during the spraying process. The accelerated cooling rate prevents YSZ or CeYSZ from timely compositional adjustments, preventing the formation of a high $Y_2O_3$-containing c (cubic) phase and low $Y_2O_3$-containing t (tetragonal) phase. It is noteworthy that the t' phase differs from the t (metastable) phase. The metastable t phase transforms to the m phase at temperatures below 873 K, whereas the t' phase is stable compared to m. With an increase in the amount of m and c phases in the coating, there is a corresponding increase in grain size within the coating material. A rise in the t' phase content not only enhances the high-temperature stability of the coating but also ensures a certain fracture toughness. Consequently, the proportion of m, c, and t' phases in the coating significantly impacts its mechanical properties, thermal stability, and thermal cyclic lifespan [37]. The distinction between the t' and t phases can be derived by calculating the tetragonality of the unit cell ($c/\sqrt{2}a$). For the t phase, $c/\sqrt{2}a > 1.010$, while for the t' phase, $c/\sqrt{2}a < 1.010$ [38]. The lattice parameters and tetragonality for YSZ and CeYSZ coatings are detailed in Table 2. An absence of $CeO_2$ phase detection in the CeYSZ coating suggests the mutual dissolution of $CeO_2$ with YSZ, resulting in the formation of a Ce, Y, and Zr solid solution. Studies have identified that the substitution of $Zr_{4+}$ by $Ce_{4+}$ can elevate the tetragonality of $ZrO_2$, subsequently enhancing the fracture toughness of the coating [25,26].

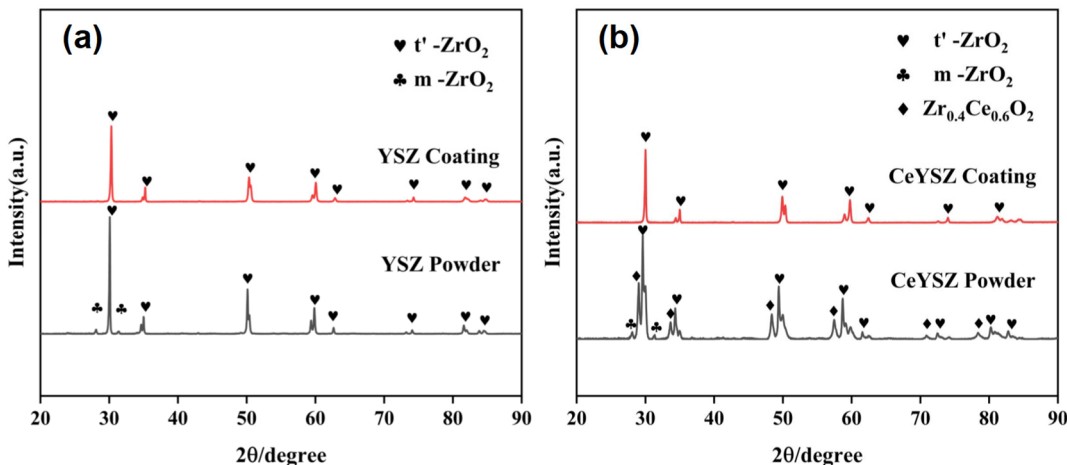

**Figure 1.** XRD spectra of ceramic powder and as-sprayed coatings. (**a**) represents YSZ and (**b**) represents CeYSZ.

**Table 2.** Lattice parameters and tetragonality results for the coatings.

| Coating Type | Lattice Parameter a | Coating Type b | Lattice Parameter c | Coating Type $c/\sqrt{2}a$ |
|---|---|---|---|---|
| YSZ | 3.607 | 3.607 | 5.152 | 1.009 |
| CeYSZ | 3.607 | 3.607 | 5.129 | 1.005 |

3.1.2. Microstructure Characterization of the Coatings

The surface morphologies and roughness of YSZ and CeYSZ coatings are depicted in Figure 2. From the provided micrographs, both coatings demonstrate regions indicative of complete powder fusion and areas of partial powder melting. Notably, droplet splatter

is evident on the CeYSZ surface, while this splattering phenomenon is more subdued in the YSZ coating. This observation suggests that certain molten CeYSZ droplets began solidifying prior to completing their spread, potentially enhancing the bond strength of the resultant coating. For both coatings, relatively smooth regions are bisected by inteconnected cracks, yielding distinct lamellar segments. The YSZ coating exhibits broader and deeper surface cracks in comparison to the CeYSZ, where a predominance of microcracks can be discerned. In certain areas, the CeYSZ exhibits more subtle crack delineations. This characteristic may imply the presence of through-thickness cracks in the YSZ cross-section, whereas CeYSZ predominantly features finer microcracks. Furthermore, the CeYSZ coating demonstrates smaller individual lamellar regions in comparison to the YSZ coating. The surface roughness for CeYSZ is measured at 3.16 μm, marginally less than YSZ's 3.46 μm. In summation, the melt state of the CeYSZ powder exhibits superior attributes when compared to that of the YSZ powder.

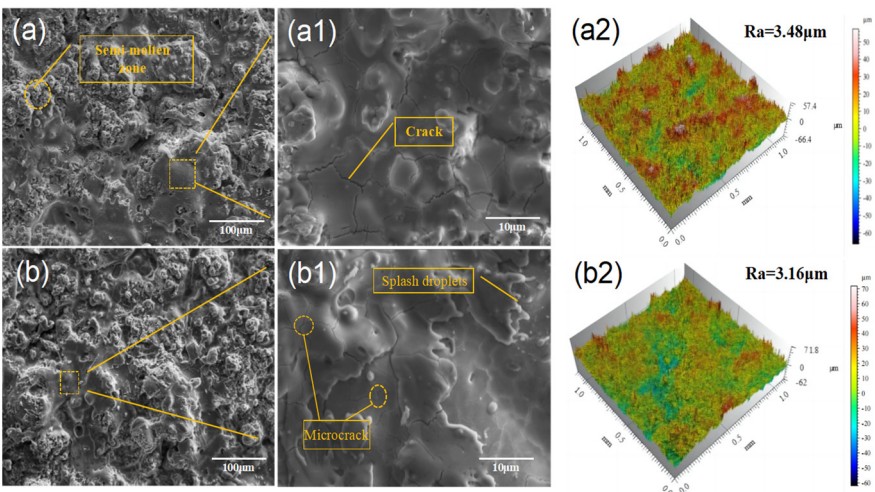

**Figure 2.** Surface microstructures of the coatings; (**a**,**b**) represent CeYSZ and YSZ, respectively, while (**a1**,**b1**) are the high magnification views of CeYSZ and YSZ, and (**a2**,**b2**) are roughness measurement results of YSZ coating and CeYSZ coating, respectively.

Figure 3 shows the cross-sectional micromorphology of the five coatings prepared by APS. The TBCs clearly exhibit the layers of the alloy substrate, metal bond coat, and the ceramic layer. These sprayed coatings present a characteristic lamellar structure. During the atmospheric plasma spraying (APS) process, powders are entrained by the carrier gas, rapidly melted in the plasma flame, and subsequently impinge onto the substrate with high velocities. As a result, the post-spray coating surface demonstrates a certain level of roughness. As depicted in Figure 3, the interfaces between the bond coat and the ceramic layer, as well as between YSZ and CeYSZ, are irregular, contributing positively to the adhesive strength of the coatings. The bond coat thickness for all these variants is approximately $100 \pm 10$ μm. The ceramic layers of TBC-1 and TBC-2 measure around $300 \pm 30$ μm in thickness. Based on the modifications during the spraying process, the bilayer ceramic structures of TBC-3, TBC-4, and TBC-5 all conform to the designed thickness ratios between the YSZ and CeYSZ layers. Elemental mapping from TBC-5 indicates a uniform distribution of Ce, Zr, and Y within the coating. With the incorporation of $CeO_2$ in the top ceramic layer of CeYSZ, the concentrations of Zr and Y are slightly reduced compared to the YSZ layer.

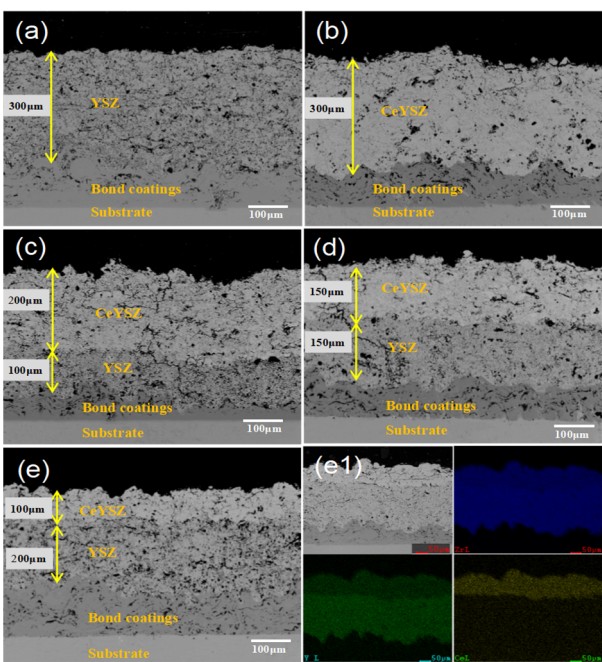

**Figure 3.** Cross-sectional microstructures of various TBCs produced by APS. (**a**) TBC-1 (YSZ), (**b**) TBC-2 (CeYSZ), (**c**) TBC-3 with a layer ratio of YSZ:CeYSZ = 1:2, (**d**) TBC-4 with a layer ratio of YSZ:CeYSZ = 1:1, (**e**) TBC-5 with a layer ratio of YSZ:CeYSZ = 2:1. (**e1**) represents the EDS (energy-dispersive spectroscopy) mapping results for TBC-5.

### 3.1.3. Porosity of the Coatings

Based on the pore distribution displayed in Figure 4 and the porosity measurements summarized in Table 3, the YSZ coating exhibited an average porosity of 13.81%, with evenly distributed pores throughout its structure. On the other hand, the CeYSZ coating showcased a lower average porosity of 9.75%, denoting a denser composition compared to the YSZ coating. A primary factor influencing this distinction is the more favorable particle size distribution of the CeYSZ powder. Under identical spraying conditions, the melting state of the CeYSZ powder is superior to that of the YSZ powder. Both coatings encompassed a mixture of large and small pores, alongside cracks. The relatively larger horizontal cracks (inter-laminar cracks) present in the YSZ coating might diminish its bonding strength. Conversely, the microcracks in the CeYSZ can mitigate the driving force for crack propagation, thus enhancing the strain toughness of the coating. The dual-ceramic layers, TBC-3, TBC-4, and TBC-5, displayed a gradient porosity, being denser at the top and sparser towards the bottom.

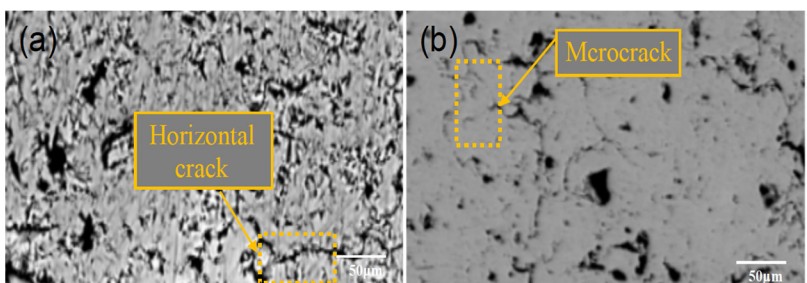

**Figure 4.** Microscopic distribution of porosity in coatings, where (**a**) represents YSZ and (**b**) showcases CeYSZ.

**Table 3.** Porosity of YSZ and CeYSZ coatings.

| Coating Type | Porosity (%) | | | | | | Average Porosity (%) |
|---|---|---|---|---|---|---|---|
| YSZ | 13.68 | 17.11 | 10.47 | 18.83 | 12.04 | 10.83 | 13.81 |
| CeYSZ | 9.26 | 10.15 | 10.74 | 9.66 | 9.58 | 9.13 | 9.75 |

*3.2. High-Temperature Oxidation Analysis of Coatings*

3.2.1. Phase Analysis of Coatings

Based on the results from Figure 5, both the YSZ and CeYSZ coatings, after being held at 1100 °C for 100 h, exclusively exhibited the t' phase. This indicates that the high-temperature oxidation tests for both coatings were conducted under conditions where no transformation from the tetragonal phase to the monoclinic phase in $ZrO_2$ occurred, effectively eliminating the influence of phase transitions on the coatings at elevated temperatures.

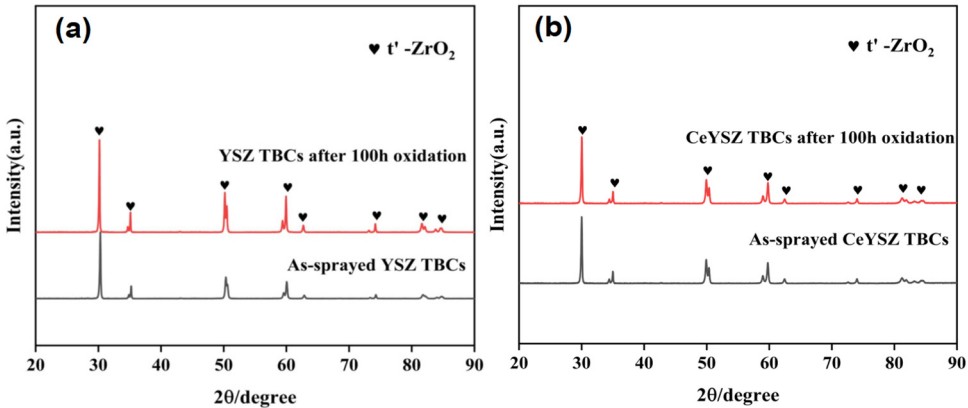

**Figure 5.** XRD spectra of the coatings after 100 h of oxidation, where (**a**) represents YSZ and (**b**) represents CeYSZ.

3.2.2. Cross-Sectional TGO Morphology and Composition after Oxidation

Figure 6 showcases the cross-sectional TGO morphology and the composition of the thermally grown oxide for the five coatings after being held at 1100 °C for 100 h. During the high-temperature oxidation process, the metallic elements in the bond coat react with oxygen in the following sequential reactions, leading to the formation of the thermally grown oxide (TGO) [35]:

$$2Al + 1.5O_2 \rightarrow Al_2O_3 \tag{1}$$

$$2Cr + 1.5O_2 \rightarrow Cr_2O_3 \tag{2}$$

$$Ni + 0.5O_2 \rightarrow NiO \tag{3}$$

$$Cr_2O_3 + NiO \rightarrow NiCr_2O_4 \tag{4}$$

As the figure illustrates, the single-layer YSZ coating exhibited sizable gaps of approximately 40–50 μm within the TGO, accompanied by cracking around these gaps. Elemental analysis of the TGO revealed an Al mass fraction of merely 1.76%, with Cr and Ni contents at 21.46% and 69.8%, respectively. This indicates the prevalent existence of oxides like $Cr_2O_3$ and NiO beyond the scant $Al_2O_3$ in the TGO. Conversely, for the single-layer CeYSZ coating oxidized under the same conditions, the TGO maintained a consistent lamellar structure with no discernible cracks. Its composition analysis revealed that the TGO was predominantly $Al_2O_3$. During the high-temperature oxidation of the coating, the elements in the bond coat engage in the oxidation reactions in the order of Al, Cr, Ni, Y [39]. Due to Al's high concentration and diffusion rate, an $Al_2O_3$ layer first forms at the interface between the bond coat and the ceramic layer. As oxidation progresses and Al is continually

depleted, the reactivity of Cr and Ni enhances, leading to the formation of $Cr_2O_3$, NiO, and $NiCr_2O_4$ via grain boundary diffusion. The emergence of these brittle spinels can induce stress concentration within the coating, precipitating premature cracking and degradation, thus impacting the coating's performance and lifespan [40–42]. For YSZ coatings, the primary cracking locations manifest within the TGO, suggesting that as Al is depleted, Cr and Ni elements continue to diffuse internally, reacting with oxygen to form unstable oxides, causing extensive TGO cracking. In contrast, in CeYSZ coatings, the bond coat's Al element is not entirely consumed, and $Al_2O_3$ remains effective in preventing further oxygen penetration into the bond coat, exhibiting superior high-temperature oxidation resistance [43]. The enhanced oxidation resistance of CeYSZ compared to YSZ might be attributed to CeYSZ's relatively dense structure, which raises the diffusion barrier for oxygen, reducing the consumption of Al in the bond coat. Additionally, the incorporation of $CeO_2$ reduces the thermal conductivity of YSZ, making CeYSZ coatings more insulating than YSZ, slowing down the oxidation of bond coat elements.

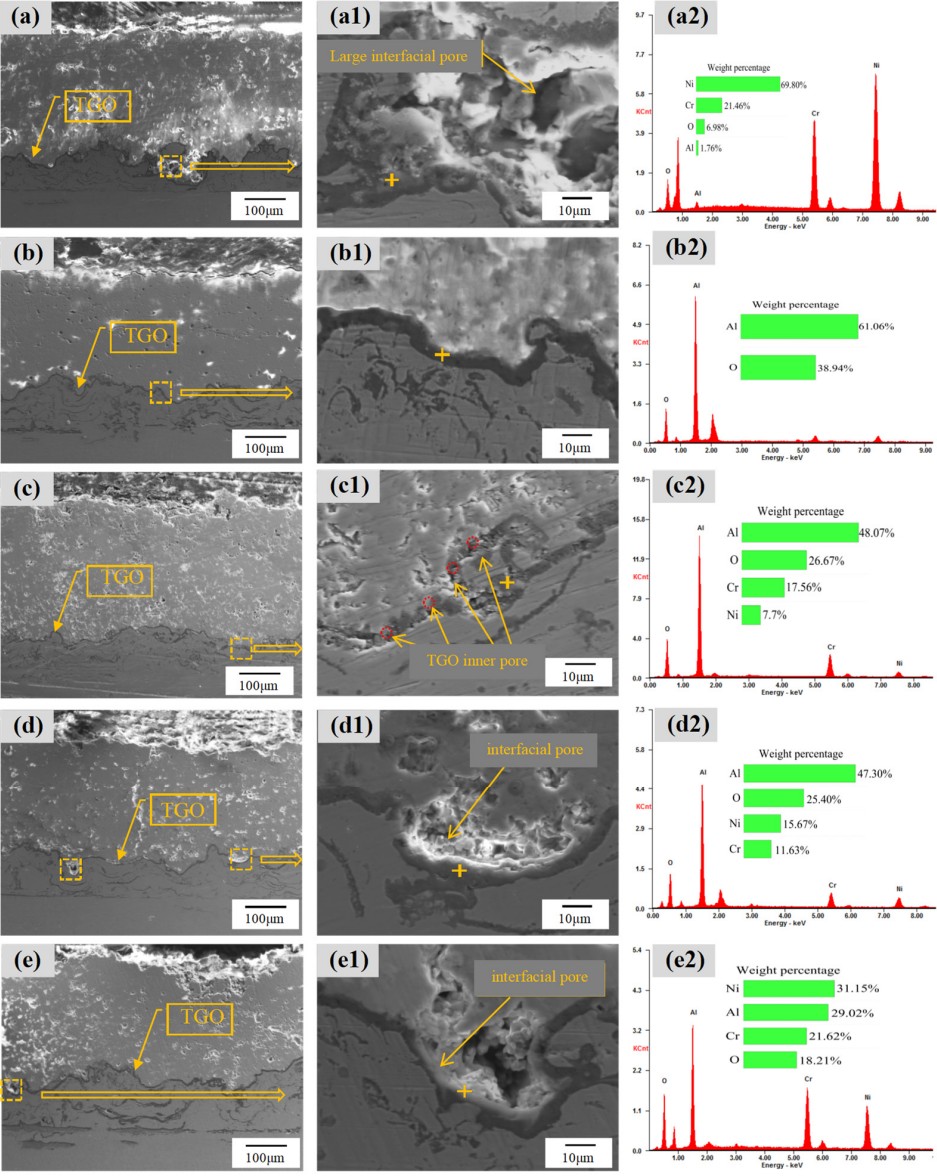

**Figure 6.** Cross-sectional scanning electron microscopy (SEM) images of (**a**) TBC-1, (**b**) TBC-2, (**c**) TBC-3, (**d**) TBC-4, and (**e**) TBC-5. The magnified micrographs from the designated rectangular areas in the main images are depicted in (**a1–e1**). Elemental spectra and compositions from the sites indicated by the "+" symbol in the enlarged images are presented in 319 (**a2–e2**).

For TBC-3, TBC-4, and TBC-5, analyzing the cross-sectional TGO morphology and composition of the thermally grown oxide reveals that after 100 h of oxidation, TBC-3 did not exhibit large-scale cracking at the interfaces between the bond coat and TGO or TGO and the ceramic layer, somewhat akin to single-layer YSZ coatings. However, there were discernible pores within the TGO, leading to granulation and localized discontinuities, rendering the TGO loosely porous overall. For TBC-4 and TBC-5, localized TGO areas displayed breakages measuring 10–20 μm and 20–30 μm, respectively—nearly halving the breakage length compared to the single-layer YSZ. In summary, after 100 h of high-temperature oxidation, the degree of TGO cracking across the coatings is as follows: TBC-1 > TBC-5 > TBC-4 > TBC-3 > TBC-2. In the dual-ceramic-layer structure, as the thickness ratio of CeYSZ increases, the Al content in the TGO rises, while the Cr and Ni contents decrease, gradually enhancing the high-temperature oxidation resistance of the coating.

From Figure 6, it is evident that the TGO cracking locations in the dual-ceramic-layered TBCs are distinctly different from those in the single-layer YSZ. The cracking in TBC-4 and TBC-5 predominantly occurs at the interface between the TGO and the ceramic layer, while in the single-layer YSZ, the cracking is localized within the TGO, with the upper boundary of the TGO remaining intact. According to the mechanisms of TGO growth [44–46], the primary reason for this difference lies in the spinel oxide formation at the interface between the TGO and the ceramic layer in TBC-4 and TBC-5. The ionic movement within these coatings is dominated by the outward diffusion of cations like $Al^{3+}$, $Cr^{3+}$, and $Ni^{2+}$. In contrast, in the single-layer YSZ coating, the ionic movement is characterized by inward diffusion of $O^{2-}$ and outward diffusion of cations. These cations react along the grain boundaries within the TGO to form unstable oxides. The growth mechanism of TGO in the single-layer YSZ leads to a thickening of the TGO, whereas in dual-ceramic-layer TBCs, such as TBC-4 and TBC-5, their growth mechanism induces internal expansion and deformation of the TGO.

### 3.3. High-Temperature Oxidation Analysis of Coatings

### 3.3.1. Phase Analysis of Coatings

The XRD patterns of the five coatings after 2 h of corrosion at 900 °C are shown in Figure 7. After thermal corrosion, all coatings underwent a phase transformation from t'-$ZrO_2$ to m-$ZrO_2$. Peaks indicative of corrosion product $YVO_4$ were observed in the YSZ coating. After corrosion, CeYSZ coating, as well as the double-ceramic structure coatings TBC-3, TBC-4, and TBC-5 with CeYSZ as the outer layer, showed diffraction peaks of both $YVO_4$ and $CeYO_4$ corrosion products, with partial precipitation of $CeO_2$. Moreover, the XRD patterns of the five coatings did not detect any mixed phases of $Na_2SO_4$ with YSZ and CeYSZ, indicating that no chemical reactions took place between the molten salt $Na_2SO_4$ and the YSZ or CeYSZ during the corrosion process.

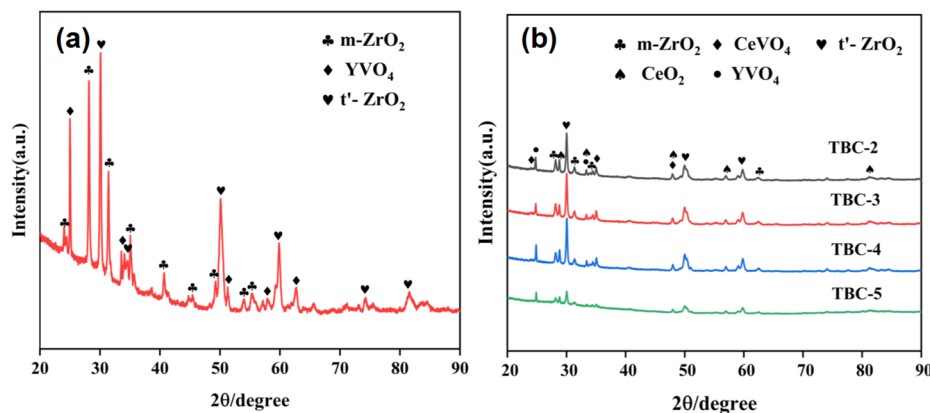

**Figure 7.** XRD patterns of the coatings after corrosion. (**a**) corresponds to TBC-1; (**b**) represents TBC-2, TBC-3, TBC-4, and TBC-5.

Analysis of the XRD spectra in Figure 7 reveals a significant phase transition of the coating within the 2θ range of (27°–33°). Peak area integration was conducted for the $(\bar{1}11)$ and (111) diffraction peaks of monoclinic $ZrO_2$, as well as the (111) diffraction peak of tetragonal $ZrO_2$ within this angular range. Using the integrated peak area as a surrogate for phase intensity, calculations were executed as per Equation (5) [47]. As shown in Figure 8, the content of m-$ZrO_2$ in YSZ was 56.1%, whereas, for CeYSZ, TBC-3, TBC-4, and TBC-5, the respective m-$ZrO_2$ contents were 30.44%, 29.01%, 23.36%, and 30.71%. Whether considering the single-layer CeYSZ or the dual-ceramic-layer structure with CeYSZ as the top layer, the post-corrosion content of the monoclinic $ZrO_2$ phase was nearly halved in comparison to the YSZ coating, indicating a reduced degree of phase transformation and exhibiting superior corrosion resistance.

$$\eta = 0.82 \times \frac{I_m(\bar{1}11) + I_m(111)}{I_m(\bar{1}11) + I_m(111) + I_{t'}(111)} \times 100\% \tag{5}$$

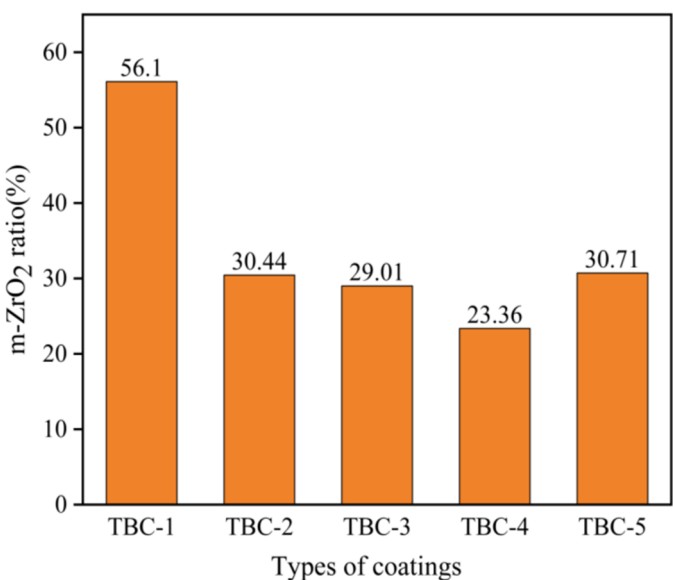

**Figure 8.** Proportion of monoclinic $ZrO_2$ phase in the five coatings after corrosion.

In the equations, $I_m(\bar{1}11)$ and $I_m(111)$ represent the peak area integration of the m-phase $ZrO_2$ diffraction peaks $(\bar{1}11)$ and (111), respectively, while $I_{t'}(111)$ corresponds to the peak area integration of the t'-phase $ZrO_2$ diffraction peak (111).

### 3.3.2. Morphology and Composition of the Coatings Post-Corrosion

Figure 9 illustrates the post-corrosion micromorphologies of the YSZ and CeYSZ coatings. The surface of the corroded YSZ coating manifested extensive corrosion product formation. These products displayed an independent rod-like structure approximately 1 μm in width, while in some localized areas, there were radiating cluster structures around 10 μm wide. These clusters primarily arise from the intertwining and stacking of the independent rod-like corrosion products. Concurrently, numerous fine particulate products have also formed on the coating surface, some of which are dispersed, while others are aggregated and distributed in clusters. In contrast, the surface morphology of the corroded CeYSZ coating, as shown in Figure 9b, markedly differs from that of the YSZ coating. After corrosion, the CeYSZ surface displayed rod-like corrosion products that consisted of multiple independent semi-cuboidal and short rod-like structures. Furthermore, other regions on the coating surface showcased a morphology where the semi-cuboidal shapes coexist with minute particulates. To further ascertain the composition of the corrosion products, energy-dispersive spectroscopy (EDS) surface mapping and line scanning analyses were conducted on both coatings.

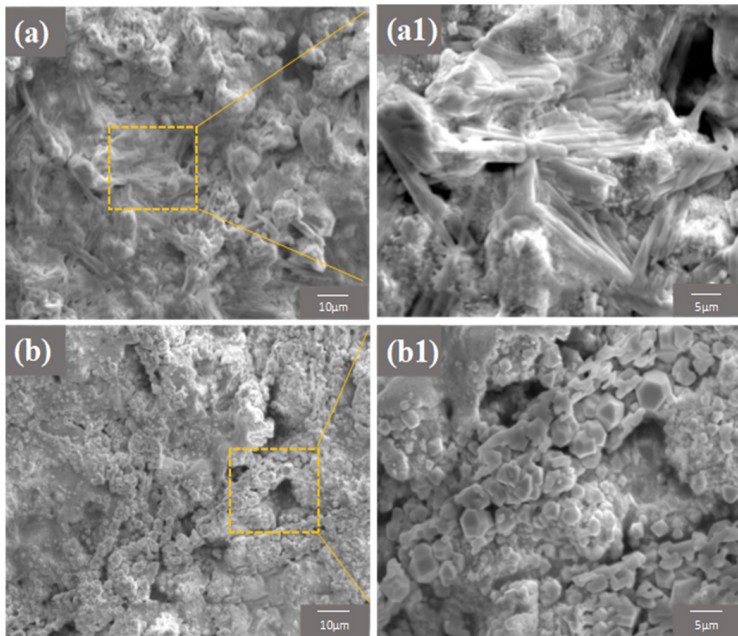

**Figure 9.** (**a**,**b**) depict the microcorrosion morphologies of the YSZ and CeYSZ coatings, respectively. (**a1**,**b1**) are magnifications of the boxed areas in (**a**,**b**).

The surface mapping results post-corrosion for the YSZ and CeYSZ coatings are presented in Figures 10 and 11. Within regions where cluster products are concentrated, shadows of Y and V elements were detected. This suggests that the corrosion products of YSZ are a Y–V compound. Combined with the XRD spectrum, this clustered product is confirmed to be $YVO_4$. Based on the EDS analysis shown in Figure 10 for CeYSZ, in areas with rod-like distributions, concentrations of Y, V, and Ce elements were simultaneously detected, indicating that the rod-like product is $CeYO_4$.

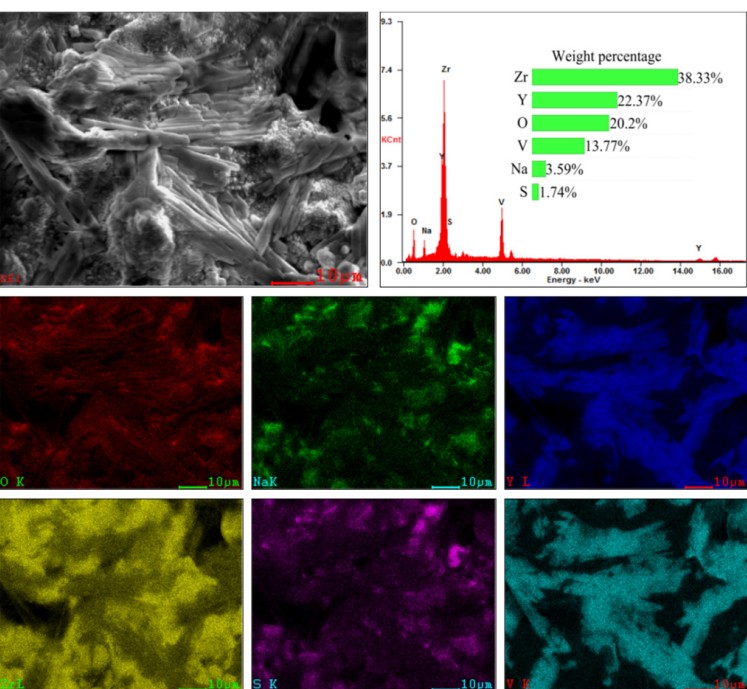

**Figure 10.** Elemental analysis of the corroded YSZ coating sample.

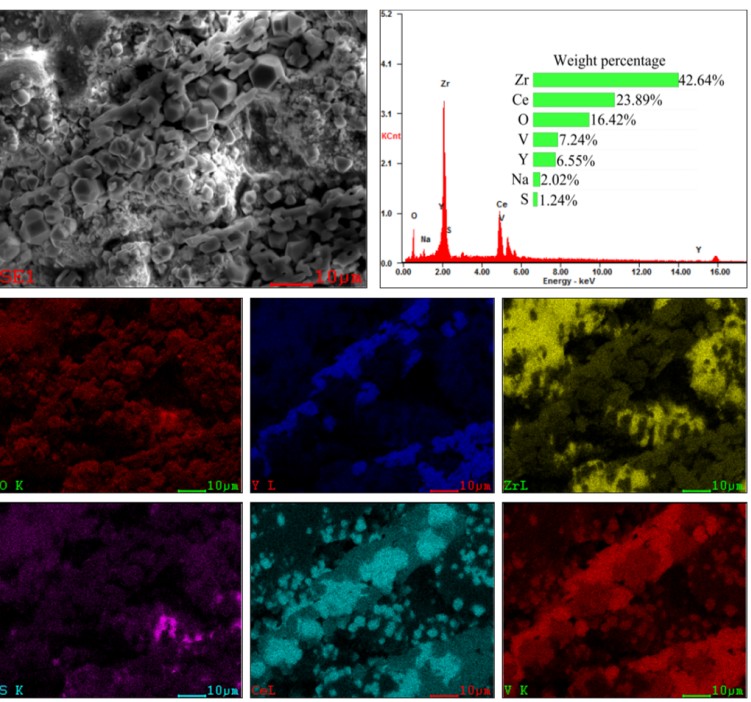

**Figure 11.** Elemental analysis of the corroded CeYSZ coating sample.

Figure 12 presents the results deduced from line scanning analyses, which aimed at verifying the compositional makeup of the spheroidal particles observed on the surfaces of YSZ and CeYSZ coatings as well as the semi-cuboidal crystals on CeYSZ. As discerned from Figure 12a, the scan line sequentially traverses a semi-cuboidal crystal, a rod-shaped crystal, another semi-cuboidal crystal, a spheroidal particle, and yet another semi-cuboidal crystal. Analyzing the scan results from Figure 12(a1), it is evident that the semi-cuboidal crystal is primarily composed of either $CeO_2$ or $CeYO_4$. Notably, apart from the Ce, there is a detectable amount of V element present at the site of the semi-cuboidal crystal. This suggests that the rod-shaped corrosion products in CeYSZ might be primarily constituted of microrod-shaped $YVO_4$, forming a backbone structure. The spaces within this framework are likely filled and intertwined with semi-cuboidal $CeO_2$ or $CeYO_4$, thus resulting in a chain-like corrosion product amalgamation of $CeO_2$, $CeYO_4$, and $YVO_4$. Within the region where spheroidal particles are densely packed between 20 and 25 μm, all elements except Zr are present in trace amounts. XRD analyses corroborate that the aggregated spheroidal corrosion products of CeYSZ post-corrosion are $m\text{-}ZrO_2$. Turning attention to the YSZ corrosion product micrographs in Figure 12b, the scan line systematically spans through a granular area, a relatively smoother surface, and then onto a smooth cluster. As deduced from Figure 12(b1), the granular region primarily consists of Zr, akin to CeYSZ, suggesting that these granules are $m\text{-}ZrO_2$. The intermediary quasi-smooth area manifests the coexistence of Zr, Y, and V. As the scan progresses towards the smooth cluster, the concentration of Zr diminishes, while Y and V exhibit an uptick. In regions where the smooth clusters dominate, the concentration of Y and V surpasses that of Zr. This leads to the deduction that the post-corrosion clustered corrosion products of YSZ coatings, primarily $YVO_4$, encapsulate minor amounts of $m\text{-}ZrO_2$. Between the coarse granular $m\text{-}ZrO_2$ region and the smooth cluster region lies a semi-smooth transition zone where $m\text{-}ZrO_2$ coexists with $YVO_4$. This observation potentially explains why post-corrosion, the YSZ coating surface manifests a markedly higher concentration of $m\text{-}ZrO_2$ compared to that of the CeYSZ coating.

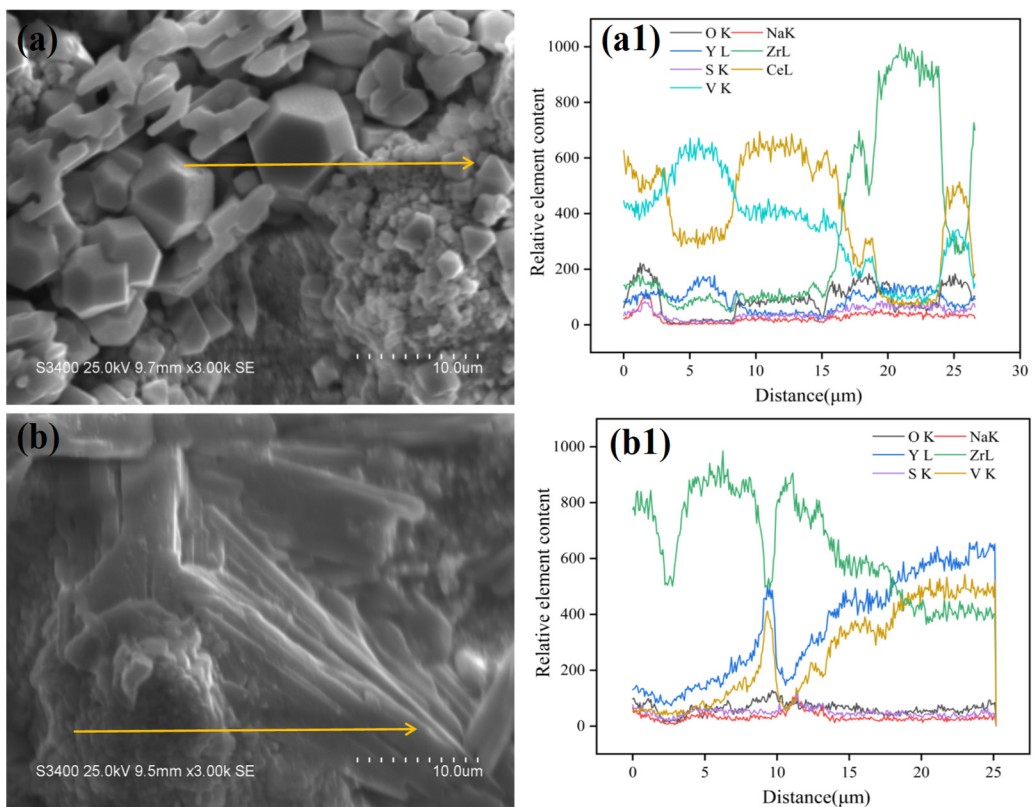

**Figure 12.** Line scanning analysis of corroded coatings. (**a**) Micrograph illustrating the corrosion products of YSZ. (**a1**) Line scan analysis corresponding to the horizontal line marked in (**a**). (**b**) Micrograph showcasing the corrosion products of YSZ. (**b1**) represents the line scan results associated with the horizontal line delineated in (**b**).

### 3.3.3. Corrosion Mechanism of the Coatings

Initially, $Na_2SO_4$ reacts with $V_2O_5$ in its molten state to produce sodium metavanadate ($NaVO_3$). Subsequently, the acidic oxide $V_2O_5$ can react with $Na_2O$ (a highly alkaline oxide), which is decomposed from $Na_2SO_4$, leading to the formation of mildly acidic sodium metavanadate ($NaVO_3$) and sodium orthovanadate ($Na_3VO_4$). The reaction processes are illustrated by Equations (6)–(8) [48,49]:

$$Na_2SO_4(l) = Na_2O(l) + SO_3(g) \tag{6}$$

$$Na_2O(l) + V_2O_5(l) = 2NaVO_3(l) \tag{7}$$

$$3Na_2O(l) + V_2O_5(l) = 2Na_3VO_4(l) \tag{8}$$

For the YSZ coating, molten $NaVO_3$ can react with the stabilizer $Y_2O_3$ to produce corrosion products $YVO_4$ and m-$ZrO_2$. Additionally, the $V_2O_5$ molten salt can also directly interact with YSZ, yielding $YVO_4$ and m-$ZrO_2$. It is noteworthy that molten $NaVO_3$ and $V_2O_5$ can easily penetrate through the porosities and cracks of the coating, depleting the stabilizer $Y_2O_3$ within the coating. This is accompanied by the destabilization of tetragonal $ZrO_2$ and the formation of monoclinic $ZrO_2$.

The specific reaction processes are illustrated in Equations (9) and (10) [48,49]:

$$ZrO_2 \cdot Y_2O_3(s) + V_2O_5(l) = m\text{-}ZrO_2(s) + 2YVO_4(s) \tag{9}$$

$$ZrO_2 \cdot Y_2O_3(s) + 2NaVO_3(l) = m\text{-}ZrO_2(s) + 2YVO_4(s) + Na_2O(l) \tag{10}$$

Regarding the CeYSZ coating, it comprises $Y_2O_3$, $CeO_2$, and $ZrO_2$. Given that $CeO_2$ is less basic than $Y_2O_3$, in accordance with the Lewis acid–base reaction theory [50], the reaction processes between the CeYSZ coating and the molten salt are depicted in Equations (11) and (12) [48,49]. $Y_2O_3$ reacts preferentially with $NaVO_3$ and not with the near-neutral $Na_3VO_4$. Due to its low basicity, $CeO_2$ does not directly interact with $Na_3VO_4$ or $NaVO_3$. However, some studies have shown that through the reaction of $NaVO_3$ and $SO_3$, $CeO_2$ can also be converted to $CeVO_4$. Furthermore, pure molten $NaVO_3$ can destabilize CeYSZ, with the resulting destabilization product being monoclinic $ZrO_2$ featuring $CeO_2$ crystals on its surface; this destabilization is attributed to a mineralization effect [48,49]. The destabilization of CeYSZ by $V_2O_5$, combined with $NaVO_3$ and $SO_3$, primarily occurs through chemical reactions that produce the corrosion product $CeVO_4$. Consequently, the main corrosion products of CeYSZ consist of a semi-rod-like (chain-like) mixture where $CeVO_4$ and $YVO_4$ are interlinked.

$$Y_2O_3(s) + 2NaVO_3(l) = 2YVO_4(s) + Na_2O(l) \tag{11}$$

$$CeO_2(s) + NaVO_3(Na_2SO_4)(l) + SO_3(g) = CeVO_4(s) \tag{12}$$

### 3.3.4. Cross-Sectional Analysis of Post-Corrosion Coatings

The cross-sectional morphologies of the five coatings after corrosion at 900 °C for 2 h are depicted in Figure 13a–d. As discerned from the images, the post-corrosion cross-sectional porosity of TBC-1 is significantly reduced, indicating a trend toward densification. In contrast, TBC-2, TBC-3, TBC-4, and TBC-5 maintain noticeable porosity post-corrosion, with their densification being less pronounced than TBC-1. This is predominantly because, after corrosion, the YSZ coating undergoes substantial phase transformations, yielding vast quantities of $m$-$ZrO_2$. The pre-existing porosity and cracks within the coatings serve as infiltration channels for the molten salts, accelerating internal corrosion. The formed corrosion product, $m$-$ZrO_2$, quickly occupies the original pores, substantially reducing the internal porosity. Concurrently, the reduction in $t'$-$ZrO_2$ content, due to densification, impairs the coating's fracture toughness.

Elemental composition analysis of the coating's cross-section near the bond coat region (A–E) reveals that the YSZ coating contains the highest molten salt element, V, at 3.68%. The CeYSZ coating has the lowest V content at 0.98%. The dual-ceramic layers, TBC-3, TBC-4, and TBC-5, have internal V contents of 1.06%, 1.58%, and 1.73%, respectively. As the thickness of the CeYSZ in the dual-ceramic layer increases from 100 μm to 200 μm, the penetration of molten salt elements is nearly halved. This suggests that, for a given ceramic layer thickness, augmenting the thickness ratio of CeYSZ to YSZ can enhance molten salt diffusion resistance, thus improving the corrosion resistance of the coating. In summary, the corrosion resistance ranking is as follows: TBC-2 > TBC-3 > TBC-4 > TBC-5 > TBC-1. The superior corrosion resistance of CeYSZ is attributed to its relatively dense structure, which impedes molten salt penetration, decelerating corrosion reactions and phase changes, thereby retaining the coating's original thermal shock resistance.

Elemental line scans of the ceramic layer for all five coatings are presented in Figure 13(a2–d2), with the scanning direction progressing from the bottom to the surface of the ceramic layer. Based on the line scan results, the YSZ coating displays a drop in Zr content around the 350 μm scan position, accompanied by an increase in Y and V content. This indicates that the primary phase in this region is the corrosion product $YVO_4$, designating it as the corroded layer [51]. Conversely, TBC-2, TBC-3, TBC-4, and TBC-5, which have CeYSZ top layers, do not exhibit such elemental shifts near the surface after corrosion, suggesting an absence of a distinct corroded layer. Following corrosion, the YSZ coating manifests a corrosion layer approximately 10 μm deep, concentrated with corrosion products. The line scan of the V element in the coatings reveals that its distribution within the YSZ region of TBC-1 and the dual-ceramic layers is uneven. High V concentrations can induce localized stress concentrations due to the growth stress of corrosion products, leading to premature coating failure.

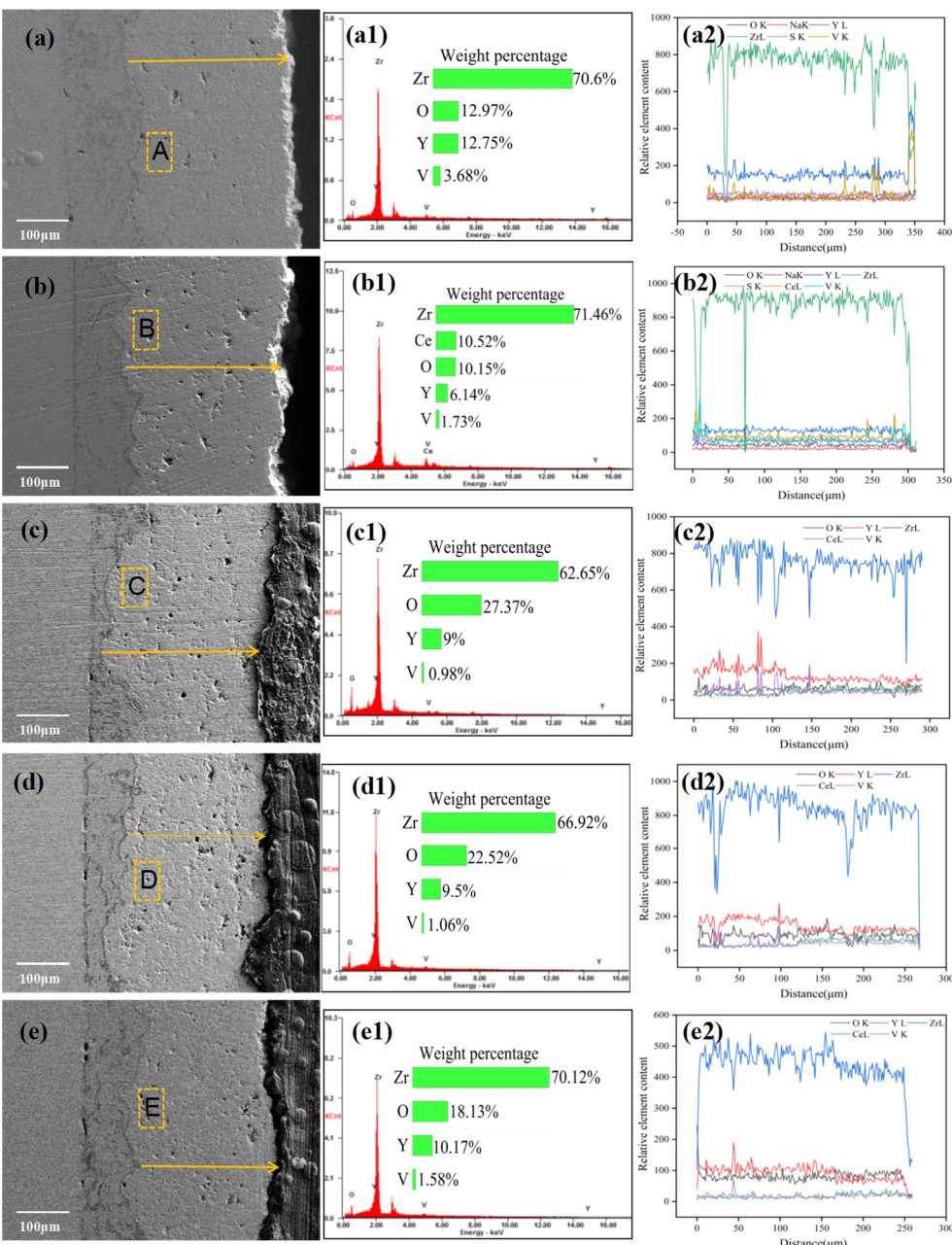

**Figure 13.** The images depict the microstructural characteristics of the corroded cross-sections for various coatings. Specifically, (**a**–**e**) represent the post-corrosion cross-sectional morphologies for TBC-1, TBC-2, TBC-3, TBC-4, and TBC-5, respectively. (**a1**–**e1**) demonstrate the elemental composition in regions A–E. (**a2**–**e2**) showcase the results of the planar scans corresponding to the horizontal lines in figures (**a**–**e**).

## 4. Conclusions

In this study, five different types of thermal barrier coatings (TBCs) were prepared via the APS method, namely TBC-1 (YSZ), TBC-2 (CeYSZ), TBC-3 (YSZ:CeYSZ = 1:2), TBC-4 (YSZ:CeYSZ = 1:1), and TBC-5 (YSZ:CeYSZ = 2:1). The key findings are as follows:

(1) Both the YSZ and CeYSZ coatings exhibited a singular t′ phase. The molten state of the CeYSZ powder demonstrated superiority over that of the YSZ powder. The sprayed CeYSZ coating, when compared to the YSZ coating, had a reduced porosity and exhibited a more dense structure.

(2) After 100 h of oxidation, the CeYSZ coating's thermally grown oxide (TGO) formed a regular band-like structure. In contrast, the TGO within the YSZ coating showed significant internal fracturing, suggesting that CeYSZ possesses superior high-temperature oxidation resistance compared to YSZ. This could largely be attributed to the relatively dense structure of the CeYSZ coating, which limits oxygen permeation, combined with the enhanced thermal insulation properties brought about by $CeO_2$ doping. Furthermore, the dual-ceramic-structured coatings TBC-3, TBC-4, and TBC-5, with increasing CeYSZ thickness, saw a shift in TGO degradation from top-layer fracturing to internal porosification. As the thickness ratio of CeYSZ to YSZ in the dual-ceramic structure is increased, the coating's resistance to high-temperature oxidation improves.

(3) The corrosion products for YSZ coatings were identified as rod-like $YVO_4$ and m-$ZrO_2$. In contrast, the CeYSZ corrosion products consisted of chain-like mixtures of $CeO_2$, $CeYO_4$, $YVO_4$, and m-$ZrO_2$. Following salt corrosion, the YSZ coating exhibited a significant phase transformation and densification, while the phase change in the CeYSZ coating was nearly halved in comparison to YSZ, retaining a certain porosity, thereby indicating commendable corrosion resistance. This can primarily be credited to the relatively dense structure of the CeYSZ, which increases the diffusion resistance to molten salts, and the doping of $CeO_2$, which reduces the consumption of stabilizers during the corrosion process.

(4) Post-salt corrosion, apart from TBC-1, which formed a corrosion layer of approximately 10 μm on its surface, no evident corrosion layers were observed on the other coatings. The distribution of the molten salt element, V, within the YSZ coating was notably non-uniform, predisposing the coating to localized stress concentrations and potential premature failures. Increasing the ratio of CeYSZ/YSZ in the dual-ceramic structure enhances the coating's corrosion resistance.

In this paper, the high-temperature oxidation and molten salt corrosion resistance of modified YSZ thermal barrier coating CeYSZ coatings and double-ceramic-layer structures are studied, and there are some limitations at the research level. In the future, the research of new thermal barrier coating materials and the innovation of coating structure should be strengthened to improve the comprehensive performance and service life of the coating.

**Author Contributions:** Conceptualization and methodology, R.L.; writing—original draft preparation, Y.X.; data curation, Q.L., Z.C., and L.G. All authors have read and agreed to the published version of the manuscript.

**Funding:** This research was funded by the Shanghai Local Capacity Building Project (21010500900), the Shanghai Engineering Research Center of Hot Manufacturing (18DZ2253400), and the National Natural Science Foundation of China (NSFC) Joint Fund Priority Program (U23A20607).

**Institutional Review Board Statement:** Not applicable.

**Informed Consent Statement:** Not applicable.

**Data Availability Statement:** Data are contained within the article.

**Conflicts of Interest:** The authors declare no conflict of interest.

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
