# Peer review of "High-Temperature Oxidation Resistance and Molten Salt Corrosion Study of YSZ, CeYSZ, and YSZ/CeYSZ Thermal Barrier Coatings by Atmospheric Plasma Spraying"

_coatings, doi:10.3390/coatings14010102_

Round 1

Reviewer 1 Report

Comments and Suggestions for Authors

The work examines one of the important problems in the field of materials science related to increasing the resistance and efficiency of materials to external influences, including corrosion and degradation at high temperatures. To protect against external influences, the authors propose the use of protective coatings based on TBC-1 (YSZ), TBC-2 (CeYSZ), TBC-3 (YSZ:CeYSZ = 1:2), TBC-4 (YSZ:CeYSZ = 1: 1), and TBC-5 (YSZ:CeYSZ = 2:1). In general, the presented work is quite promising and interesting, and the results obtained have a high level of scientific novelty and practical significance. This work can be accepted for publication, as it corresponds to the topic of the journal, but before accepting it, the authors should answer a number of questions from the reviewer that arose while reading it.

1. The authors should explain the presence of corrosion products in the form of chromium and nickel oxides, in particular, the nature of their appearance and concentration in the coating composition. How exactly these oxides are formed, and whether the type of coating affects their concentration.

2. The authors should reflect the results of mapping lateral chips along with spectra or ratios of elements, which will make it possible to determine exactly what degradation mechanisms are occurring.

3. The authors should explain what the concentration of oxide inclusions is, according to X-ray phase analysis data, and whether it changes.

4. What types of cracks form on the surface of coating samples, as well as how exactly they propagate in the samples.

5. On what basis was the concentration of impurity inclusions in the form of m-ZrO2 calculated using formula (1)?

Author Response

Dear reviewer, I have made the following changes and responses to the suggestions you made:

1Based on the review of relevant literature and the analysis of the ratio of each element in TGO, the Cr and Ni oxides can be identified as Cr2O3, NiO, and NiCr2O4, and the relevant reaction equations are added. Because of the existence of a certain porosity of the plasma sprayed coating, the oxygen in the air through the pores of the coating and the Cr and Ni in the metal bonding layer can react to generate the spinel oxides mentioned above. Due to the poor electrical conductivity of the ceramic material, the specific morphology of the spinel oxide could not be observed. Since, spinel oxide has a large growth stress and the elemental content of Cr and Ni in TGO is higher. Therefore, the concentration of Cr2O3, NiO, and NiCr2O4 can be reflected by the morphology and elemental composition of TGO.

2Cr + 1.5O2 → Cr2O3                                                                         

Ni + 0.5O2 → NiO                                                                            

Cr2O3 + NiO → NiCr2O4 

From the results of this paper, it is concluded that the material and structural design of the coatings are closely related to their concentrations.The relatively dense structure of CeYSZ increases the diffusion resistance of oxygen within the coating and reduces the consumption of Al elements in the bonding layer.The doping of CeO2 reduces the thermal conductivity of YSZ, which makes the thermal insulation of the CeYSZ coating superior to that of the YSZ coating, thus slowing down the oxidation of the elements in the bonding layer. Therefore, the concentration of Cr2O3, NiO,and NiCr2O4 in TGO is high after 100h of oxidation of YSZ coating, and the TGO composition of CeYSZ coating is still Al2O3.In the double ceramic layer structure, as the thickness ratio of the CeYSZ layer becomes bigger, the content of Al in TGO rises, the content of Cr and Ni decreases, and the coating's resistance to high-temperature oxidation is gradually improved.

2According to the suggestions made by the reviewer, the tables of elemental contents in Figures 6, 10, 11 and 13 have been changed into the form of bar charts, which can reflect the changes in the contents of each element more intuitively.

3The XRD patterns in the high-temperature oxidation experiments were mainly used to analyse the phase transformation of the two coatings after being held at 1100°C for 100 h. After comparative XRD analyses and the accumulation of literature, the two coatings have good phase stability at this temperature. Therefore, the influence of the phase transformation or the change of oxide concentration on the high temperature corrosion resistance of the coatings was excluded.

4Thermal stress and TGO growth stress usually cause the coating to form cracks at the interface between the ceramic layer and the bonding layer, and with the accumulation of thermal stress, the cracks in the coating gradually grow and intersect into penetrating cracks, which in turn generate surface cracks. In this experiment, the cracks on the surface of the coating after the high temperature oxidation experiment are in the form of micro-network cracks. The best experimental method to study cracks is the thermal shock water quenching experiment of the coating, and since this paper focuses on the static high-temperature oxidation experiment, the cracking condition of the coating is not analysed.

5Equation (5) is the reference formula for calculating the phase transition of the coating in a large number of literatures. Analysing the XRD patterns, it can be obtained that the phase change of the coating is obvious in the range of 2θ of (27°-33°), and the peak area integrations are performed for the (11) and (111) diffraction peaks of monoclinic ZrO2 and (111) diffraction peaks of tetragonal phase ZrO2 in this angular range. By replacing the physical phase intensity with the diffraction peak area integral, the degree of phase change of the coating can be accurately derived by calculating according to the above formula. In order to make the article more concise, the detailed calculation procedure is not given in the paper.

Finally, once again, I would like to express my most sincere thanks to my teacher for the valuable advice he provided.

Reviewer 2 Report

Comments and Suggestions for Authors

The authors provided interesting results on thermal barrier coatings. Below are a few comments to be addressed.

1. Add expansion of APS in the abstract.

2. Why authors have selected the Air Plasma Spray method - this can be mentioned in the introduction.

3. There are two Table 2. In the second Table 2, the term coating type is used repeatedly and confusingly.

4. Figure 2 caption is not complete.

5. The details of porosity measurements shall be added in the experimental part.

6. Equations need to be numbered

7. The different areas of the sem image in Figure 13e shall be labelled/named.

Author Response

Dear reviewer, I have made the following changes and responses to the suggestions you made:

1、The abstract adds an extension of APS, namely atmospheric plasma spraying ( APS).

2、This paper does need to elaborate on the advantages of the APS method of coating preparation. To this end, the following additions are made: common coating preparation techniques are atmospheric plasma spraying (APS), electron beam physical vapour deposition (EB-PVD), high-speed flame spraying (HVOF). Plasma spraying is widely used in the repair of thermal barrier coatings, wear-resistant coatings, corrosion-resistant coatings, and the near-accurate moulding of fine ceramics due to the advantages of simple operation, low cost, fast preparation speed, high deposition efficiency, etc. It is especially suitable for the preparation of thick coatings (300-3000 μm). Therefore, all the coatings in this paper are prepared by the APS method.

3、The first Table 2 has been changed.

4、The title of Fig. 2 has been completed.

5、The details of porosity measurement have been added in the experimental section. Six areas were randomly selected in the cross-section of the coating, and the images were processed in grey scale with the help of image J software to calculate the ratio of the pore area to the area as the porosity of the coating, and the average value of multiple measurements was recorded as the average porosity of the coating.

6、The equations have been numbered.

7、The results of the elemental surface scanning in Fig. 13e for the five regions of areas A, B, C, D and E are labelled in the corresponding areas in Fig. 13. In order to more clearly describe the proportions of each element, the elemental compositions and occupancies are presented in the form of bar charts.

Finally, once again, I would like to express my most sincere thanks to my teacher for the valuable advice he provided.

Reviewer 3 Report

Comments and Suggestions for Authors

The work is devoted to an important topic. Methodologically, it is performed at a high level, but there are certain shortcomings in the analysis of the results obtained:

1. Raises doubts about the use of the said melt (а mixture of Na₂SO₄ and V₂O₅ in a 50%:50% weight ratio) for corrosion testing. At the same time, the reference [28] speaks of the use of a mixture of salts.

2. In the "Introduction" section, it is necessary to pay attention not so much to the phase composition of the coatings, as to the thermal stability of the applied layers. This will determine their performance. Therefore, I believe that it is necessary to analyze the pain of a wide range of publications. In particular, it is necessary to refer to the works of the others authors, for example, https://doi.org/10.3390/app122211617.

3. It is necessary to clearly formulate the purpose of the research, which should correlate with the conclusions of the work.

4. Determination of the porosity of the microstructure from a photo without statistical analysis or without the use of known techniques is also questionable.

 After corrections, additions and reasoned explanations, the article can be published.

Author Response

Dear reviewer, I have made the following changes and responses to the suggestions you made:

1、Molten salt corrosion of thermal barrier coatings is a form of coating failure. Most of the fuels for gas turbines contain elements such as sulphur and vanadium, which react with air to produce corrosive salts such as V2O5 and Na2SO4. Most of the conventional molten salt corrosion experiments use a mixture of V2O5 and Na2SO4 salts, with some literature references. These salts can penetrate the coating after turning into molten salts at high temperatures and react with the stabiliser of the coating. The phase transition of the reaction process can lead to the concentration of thermal stresses within the coating, inducing premature coating failure. Therefore, there is a theoretical basis for using a mixture of (Na₂SO₄ and V₂O₅ with a weight ratio of 50%:50%) for corrosion testing.

2、It is really important to pay attention to the phase composition of the coating, and according to the comments made by the teacher, the phase composition and the phase transition process of the YSZ coating are explained in the introduction. t' phase (sub-stable tetragonal phase) present in the YSZ coating can be stabilised at 1200°C, but when the service temperature of YSZ exceeds 1473 K, the transition from t' phase to t (tetragonal phase) and c (cubic phase) will take place, and on cooling the t-phase transforms to the m-phase (monoclinic phase), which is accompanied by a volume expansion of 3-5% and leads to cracking of the coating.

3、The purpose of the study is closely related to the conclusion, and according to the reviewer's opinion, the purpose of the study is further clarified at the end of the introduction. Because it was mentioned earlier that the growth stress of high temperature oxidation TGO and molten salt corrosion are important causes of coating failure, the main purpose of this paper is to analyse the TGO growth and damage mechanism and corrosion behaviour and mechanism of YSZ and CeYSZ coatings, compare the resistance to molten salt corrosion and high temperature oxidation of the two types of coatings, and investigate the effect of the ratio of thickness of the two types of coatings in the dual-ceramic layer structure on the overall performance of the coatings. The influence of the thickness ratio of the two coatings in the double ceramic layer structure on the comprehensive performance of the coating is also investigated.

4、6 areas were randomly selected in the cross-section of the coatings, and the images were processed in grey scale with the help of image J software to calculate the ratio of the pore area to the area as the porosity of the coatings, and the average value of multiple measurements was recorded as the average porosity of the coatings. Most of the conventional porosity tests of thermal barrier coatings are calculated by adjusting the grey scale with the help of image processing software. However, there may be some errors in the measurement results, this paper focuses on testing and comparing the porosity of YSZ coating and CeYSZ coating, and the porosity of the two coatings are tested under the same image grey scale conditions, so the difference between the two porosities has a small error. Checking the relevant references, it is known that the porosity of plasma sprayed YSZ thermal barrier coatings is between 10% and 20%, and the porosity varies with different process parameters of spraying. In this paper, the porosity of the YSZ coating is measured to be 13.81%, which is in line with the above range.

Finally, once again, I would like to express my most sincere thanks to my teacher for the valuable advice he provided.

Reviewer 4 Report

Comments and Suggestions for Authors

In general, the scope of the paper is interesting; however, a few issues need to be addressed by the authors. The first generic comment is that the references should be linked with the corresponding number. Furthermore, the manuscript needs to be proofread in terms of English Grammar/Syntax, and better formatting between paragraphs is required.  You will find the comments below which are divided according to the chapters of the manuscript.

1.       Introduction

·       The introduction section includes the literature review. These two sections should be divided.

·       More information about the selected material of coating should be provided.

·       More information about the APS method should be provided.

·       Row 63: Passive voice should be used.

2.       Materials & Methods

2.1.   Materials

·       Row 76: There are extra symbols [).] in this row. Also, the abbreviation APS has been set with different words above.

·       What spraying gun is used? What is the motion system used for APS?

2.2.   High-Temperature Oxidation Experiment

·       Row 89: By which method are samples sectioned, grinded, and polished?

2.3.   Molten Salt Corrosion Test

·       Row 93: How is the mixture applied in the coating surface with this accuracy of material?

3.       Results and Discussion

3.1.   Characterization of As-Sprayed Coatings

·       Figure 1 should have a better analysis. This comment also refers to the other diagrams.

·       Row 165: The verb in this sentence is missing.

·       What about the required surface roughness and the required dimensional accuracy of the coating? Does this coating need extra post-processing to meet these requirements?

3.2.   -

3.3.   High-Temperature Oxidation Analysis of Coatings

·       Row 304: Equation (1) should be introduced properly and not as a figure.

4.       Conclusions

The issue here is that the work should be aimed at a broader objective, as it is presented in a stand-alone way. For example, all the output data for microstructure characterization should be correlated with process inputs or with a failure mechanism in order to be put into a manufacturing framework for APS. So, understanding the size of the work, it should be directed towards some overall framework that would be presented in future work.

The literature review should be enhanced with additional material and more recent material as it is very limited. Some indicative papers follow:

·       P. Stavropoulos, H. Bikas, T. Bekiaris, "Combining process and machine modelling: A Cold Spray Additive Manufacturing case", 20th CIRP Conference on Electro Physical and Chemical Machining, Volume 95, pg. 1015-1020 , 19-21 January, Zurich, Switzerland (2021)

·       Vaßen, Robert, et al. “Unique Performance of Thermal Barrier Coatings Made of Yttria‐Stabilized Zirconia at Extreme Temperatures (>1500°C).” Journal of the American Ceramic Society, vol. 104, no. 1, 22 Sept. 2020, pp. 463–471, https://doi.org/10.1111/jace.17452. Accessed 8 Dec. 2023.

Comments on the Quality of English Language

In general, the scope of the paper is interesting; however, a few issues need to be addressed by the authors. The first generic comment is that the references should be linked with the corresponding number. Furthermore, the manuscript needs to be proofread in terms of English Grammar/Syntax, and better formatting between paragraphs is required.  You will find the comments below which are divided according to the chapters of the manuscript.

1.       Introduction

·       The introduction section includes the literature review. These two sections should be divided.

·       More information about the selected material of coating should be provided.

·       More information about the APS method should be provided.

·       Row 63: Passive voice should be used.

2.       Materials & Methods

2.1.   Materials

·       Row 76: There are extra symbols [).] in this row. Also, the abbreviation APS has been set with different words above.

·       What spraying gun is used? What is the motion system used for APS?

2.2.   High-Temperature Oxidation Experiment

·       Row 89: By which method are samples sectioned, grinded, and polished?

2.3.   Molten Salt Corrosion Test

·       Row 93: How is the mixture applied in the coating surface with this accuracy of material?

3.       Results and Discussion

3.1.   Characterization of As-Sprayed Coatings

·       Figure 1 should have a better analysis. This comment also refers to the other diagrams.

·       Row 165: The verb in this sentence is missing.

·       What about the required surface roughness and the required dimensional accuracy of the coating? Does this coating need extra post-processing to meet these requirements?

3.2.   -

3.3.   High-Temperature Oxidation Analysis of Coatings

·       Row 304: Equation (1) should be introduced properly and not as a figure.

4.       Conclusions

The issue here is that the work should be aimed at a broader objective, as it is presented in a stand-alone way. For example, all the output data for microstructure characterization should be correlated with process inputs or with a failure mechanism in order to be put into a manufacturing framework for APS. So, understanding the size of the work, it should be directed towards some overall framework that would be presented in future work.

The literature review should be enhanced with additional material and more recent material as it is very limited. Some indicative papers follow:

·       P. Stavropoulos, H. Bikas, T. Bekiaris, "Combining process and machine modelling: A Cold Spray Additive Manufacturing case", 20th CIRP Conference on Electro Physical and Chemical Machining, Volume 95, pg. 1015-1020 , 19-21 January, Zurich, Switzerland (2021)

·       Vaßen, Robert, et al. “Unique Performance of Thermal Barrier Coatings Made of Yttria‐Stabilized Zirconia at Extreme Temperatures (>1500°C).” Journal of the American Ceramic Society, vol. 104, no. 1, 22 Sept. 2020, pp. 463–471, https://doi.org/10.1111/jace.17452. Accessed 8 Dec. 2023.

Author Response

Dear reviewer, I have made the following changes and responses to the suggestions you made:

1、The introductory section was reorganized based on the teacher's comments.

2、The characterization of the coating material YSZ, in particular the phase transition process, was added.Yttria-stabilized zirconia (YSZ), as the most classic TBC material, hosts the t' phase (metastable tetragonal phase) in the coating, which remains stable at temperatures up to 1200°C. This stability enhances the coating's strain tolerance and resistance to thermal shock. Due to its exceptional thermophysical and mechanical properties, YSZ is considered the most widely used TBC material to date. However, when the operating temperature of YSZ exceeds 1473 K, a transition occurs from the t' phase to the t phase (tetragonal phase) and c phase (cubic phase). During cooling, the t phase transforms into the m phase (monoclinic phase). This phase transformation is accompanied by a 3-5% volume expansion, leading to the formation of cracks in the coating.

3、Information about APS is added in the introductory section.Common coating preparation techniques include atmospheric plasma spraying (APS), electron beam physical vapor deposition (EB-PVD), and high-velocity oxygen fuel (HVOF) spraying. Atmospheric plasma spraying, due to its simplicity, low cost, rapid preparation speed, and high deposition efficiency, is widely used in the fabrication of thermal barrier coatings, wear-resistant coatings, restoration of corrosion-resistant coatings, and near-net shaping of advanced ceramics. It is particularly suitable for the preparation of thick coatings (300-3000 μm).

4、Passive voice has been changed

5、The symbols at line 76 correspond to the parentheses in the previous line. the full name of the APS is changed to atmospheric plasma spraying. the motion system of the atmospheric plasma spraying equipment is performed by an ABB robot, and the model of the spray gun is F4-MB 6mm.

6、In the high-temperature oxidation experiments, the five coating specimens were cut with a grinding wheel sheet, the specimens were heat-immersed with epoxy resin, and finally the samples were sanded and polished with SiC sandpaper with 2.5 μm polishing paste.

7、In the molten salt corrosion experiments, the salt mixture was uniformly applied to the surface of the coatings at a rate of 10 mg /cm2 . Since the coated specimens were 25.4 mm diameter discs, each coated specimen had the same surface area of 5.06 cm2. The mass of corrosive salt for each coated specimen was calculated to be 50.6 mg, which was weighed with a high-precision balance, and then uniformly applied to the surface of the coatings. The corrosive salt was applied at a distance of 2-3 mm from the edge of the coating.

8、Figure 1 mainly analyzes a phase transition process of the two materials from the powder state to the coating state, and the tetragonal degrees of the two coatings in Table 2 are measured directly from the Jade software.

9、Additional changes have been made to the figure name for figure 2. Surface Microstructures of the Coatings, (a) and (b) represent CeYSZ and YSZ respectively, while (a1) and (b1) are the high magnification views of CeYSZ and YSZ,(a2) and (b2) are roughness measurement results of YSZ coating and CeYSZ coating respectively.

10、The roughness of the coating was measured using a Smart proof 5 confocal microscope and the average of 10 measurements was taken as the surface roughness value of the coating. The experimental values are in the micron range and no treatment of the coating was required before or after the measurement.

11、The original formula section has been re-edited.

12、The teacher's comments on the conclusion section were very valuable. For this reason, the limitations and future development objectives of the paper are pointed out in the conclusion. In this paper, the high temperature oxidation and molten salt corrosion resistance of modified YSZ thermal barrier coatings CeYSZ coatings and dual ceramic layer structures were investigated, and there are some limitations at the research level. In the future, the research on new thermal barrier coating materials and the innovation of coating structure should be strengthened to improve the comprehensive performance and service life of the coatings.

The references section was linked to the original text according to the teacher's suggestion to increase the number of references in the article. These references provided to me by my teacher will be of great help to my future studies. Finally, once again, I would like to express my most sincere thanks to my teacher for the valuable advice he provided.

Round 2

Reviewer 1 Report

Comments and Suggestions for Authors

The authors answered all the questions posed, the article can be accepted for publication.

Author Response

Thank you very much for your valuable advice, which is of great help to me. Kind regards.

Reviewer 3 Report

Comments and Suggestions for Authors

After additions made by the authors and reasoned responses to comments, the article looks better and can be recommended for publication

Author Response

(The authors gave the same response as above.)
